# Self-Efficacy as a Moderator between Stress and Professional Burnout in Firefighters

**DOI:** 10.3390/ijerph16020183

**Published:** 2019-01-10

**Authors:** Marta Makara-Studzińska, Krystyna Golonka, Bernadetta Izydorczyk

**Affiliations:** 1Department of Health Psychology, Faculty of Health Sciences, Jagiellonian University Collegium Medicum, Kopernika 25, 31-501 Kraków, Poland; marta.makara-studzinska@uj.edu.pl; 2Institute of Applied Psychology, Faculty of Management and Social Communication, Jagiellonian University, Łojasiewicza 4, 30-348 Kraków, Poland; bernadetta.izydorczyk@uj.edu.pl

**Keywords:** burnout, perceived stress, self-efficacy, firefighters

## Abstract

The purpose of the study is to analyze the importance of individual resources in firefighting, one of the highest risk professions. Firefighters from 12 different Polish provinces (*N* = 580; men; M (mean age) = 35.26 year, SD = 6.74) were analyzed regarding the perceived stress at work, burnout, self-efficacy, and a broad range of sociodemographic variables. The Perceived Stress Scale (PSS), the Link Burnout Questionnaire (LBQ), and the General Self-Efficacy Scale (GSES) were used in the study. To explore the relationships between work-related stress, burnout, and self-efficacy, separate regression models for each burnout dimension were analyzed. The results revealed that self-efficacy is a significant moderator that changes the direction and strength of the relationships between perceived stress and psychophysical exhaustion, sense of professional inefficacy, and disillusion. However, self-efficacy did not moderate the relationship between stress and lack of engagement in relationships (relationship deterioration). The results indicate that self-efficacy in firefighters is a crucial personal resource that buffers the impact of perceived stress on most burnout symptoms. It may be concluded that in high risk professions, special attention should be paid to developing self-efficacy as an important part of burnout prevention programs, pro-health activities, and psychoeducation.

## 1. Introduction

Firefighting is one of the highest risk professions in which repetitive exposure to traumatic experiences may lead to serious psychological consequences [1,2,3,4,5,6,7,8,9,10,11,12,13,14,15,16,17]. Current studies indicate that the specific stress and workload of professional firefighters increases the risk of mental health symptoms [1], post-traumatic stress disorder [2,3], burnout [4,5,6,7,8], depression [9], and suicidality [10,11,12]. In this context, researchers accentuate the role of individual characteristics that may influence the potential consequences of experienced stress [10,13,14,15,16,17].

In search of psychological moderators that affect the relationship between stress and occupational burnout in people who regularly experience traumatic events at work, it is worth emphasizing the role of self-efficacy—a personal resource which is a component of self-structure. In the literature, self-efficacy is defined as one’s belief of the possibility of achieving an intended goal in a specific life situation [18,19,20,21]. Perceived self-efficacy is a psychological variable believed to be important for motivation to take action and to be supportive for the effectiveness of various undertaken activities [22]. Self-efficacy is a conviction that one is able to carry out a certain activity and get a successful result. 

Self-efficacy as a personal resource that affects human behavior appears in different research areas such as health psychology [23], personality [24], social psychology [18,25], and more often in work psychology [19,26,27]. 

In the context of work, the level of self-efficacy allows the ability to cope in a given professional situation to be defined. The sense of self-efficacy is considered to be an important mechanism of human behavior self-regulation, differentiating people as to their cognitive and motivational functioning and influencing employee behavior in organizational environments. Self-efficacy supports employees’ mental well-being and positive coping with stress [26]; it is also a crucial component of self that supports demanding actions that require persistence and are characterized by a high level of complexity [19]. 

In the area of work psychology, the significance of self-efficacy is emphasized due to the problem of burnout syndrome [28,29,30,31,32,33,34,35]. Our literature review shows that burnout syndrome is an important and developing research area in which many studies explore its antecedents [36,37] and consequences [38,39,40,41]. The most influential burnout model developed by Maslach and colleagues [40] is described by three dimensions: exhaustion, depersonalization/cynicism, and lack of personal accomplishment. Exhaustion evokes emotional and cognitive distancing from one’s work, and depersonalization/cynicism from relation with others. Maslach et al. [40] show that the lack of personal accomplishment is in complex relation to other burnout dimensions, which may be a function of exhaustion and/or cynicism. Exhaustion and cynicism may interfere with personal accomplishment and all three dimensions may develop parallelly. Research focuses on the sociodemographic and psychosocial determinants of occupational burnout in many professions, including paramedics [42], nurses [43], doctors [44], teachers [28,30,31,45], firefighters [5,27], and other intervening and caring professions [46,47]. 

In this work we concentrate on fire brigade officers, taking into account the broad spectrum of socioeconomic indicators and psychological characteristics that may be important for self-efficacy as a moderator between professional stress and burnout. Studies exploring the strength of the influence of self-efficacy as a moderator variable between stress and burnout in firefighters is quite a niche research area. Although the relationship between self-efficacy and personal accomplishment seems to be predictable, exploring the relationships between self-efficacy and other burnout dimensions may broaden the knowledge of the role of self-efficacy in developing burnout syndrome in high risk professions of firefighters. Additionally, the scope of the present study is to investigate two different aspects of self-efficacy: 1) a general belief in one’s effectiveness in dealing with hurdles and 2) professional efficacy that refers to evaluation of one’s own professional competences. Prati et al. [16] revealed that perceived stress among rescue workers (firefighters, paramedics, and medical technicians) was significantly related to professional quality of life, but only in a group with a low level of self-efficacy. In subjects with higher levels of self-efficacy, no significant relation was observed. Thus, in this group of rescue workers self-efficacy acted as a buffer between perceived stressful events at work and quality of life. This result suggests that this personal resource in firefighters may play a significant role when considering burnout as a consequence of work-related stress. Due to high occupational stress, firefighters are susceptible to frequent and severe damage to their mental and physical health because, when saving lives, they experience various risk factors in their work environment on a daily basis [27,48].

It can be assumed that firefighters function in a state of persistent tension—constantly being ready to be called out to perform rescue operations in their work environment. In everyday work, a firefighter encounters tasks that can be regarded as struggling with life-threatening situations. They face traumatic events that evoke a natural reaction of fear, dread, helplessness and diverse coping strategies. Firefighters’ perception of and involvement in the emotional states of victims may result in the development of stress that is defined as a “secondary traumatization” [49]. On the other hand, firefighters’ personal resources can be a protective factor that supports constructive strategies for coping with stress; over time, they can lead to the development of so-called vicarious post-traumatic growth (VPTG) [50]. The way in which firefighters’ professional careers, coping strategies, or occupational burnout develop depend, among other factors, on personal resources. As self-efficacy may play a crucial role as a protective factor in preventing stress and burnout, it is particularly important to explore and extend this research area. 

The aim of the present study was to determine whether firefighters’ self-efficacy may be an important moderator between occupational stress and burnout. Presented research investigations have been undertaken to determine whether high self-efficacy correlates negatively with occupational stress and occupational burnout.

The assessment of the impact of self-efficacy in the development of professional firefighters’ burnout may be of the great importance for shaping positive employee attitudes, proper human resource management and for increasing the retention of specialists in emergency services. Identifying self-efficacy as psychological factor that moderates between stress and burnout also seems to be important for the design and planning of intervention programs for professional firefighters.

Due to the theoretical background, indicating a possible protective effect of resources in the stress-burnout relationship, an analysis of intermediary variables was carried out. We hypothesize that self-efficacy moderates the relationship between occupational stress and burnout in a group of firefighters.

The following research questions were formulated:
1)How does self-efficacy moderate the relationship between occupational stress and burnout in a group of firefighters?2)What is the extent of self-efficacy’s moderation of the relation between stress and specific dimensions of burnout, i.e., psychophysical exhaustion, deterioration in relations with others, lack of self-efficacy, and disappointment among firefighters?

## 2. Materials and Methods

In the study, the dependent variables were burnout indicators, the predictor was perceived stress, and the moderating variable was self-efficacy. The analyzes were carried out separately for each of the burnout indicators as a dependent variable. 

The variables are operationalized as follows:
1)Occupational burnout refers to subjective feelings about one’s job in relation to the level of psychophysical exhaustion, deterioration in relations with a person who is professionally assisted (e.g., an injured person or victim), lack of professional efficacy, and the level of disappointment (disillusion) among firefighters.2)Perceived occupational stress describes a subjective evaluation of experienced stress relating to one’s actual situation in a professional context.3)Sense of self-efficacy denotes a subjective appraisal of belief in one’s effective coping with difficult situations and obstacles.

The following sociodemographic variables were controlled for in the presented study: age, gender (due to the availability of the group of respondents and the specificity of this profession, only men were examined), marital status, place of residence (province, village, city), education, work experience, and professional status.

Additionally, following sociodemographic variables were controlled for in the presented study: age, gender (due to the availability of the group of respondents and the specificity of this profession, only men were examined), marital status, place of residence (province, village, city), education, work experience, and professional status.

### 2.1. Participants

Purposive sampling was used to select respondents from fire brigades involved in the rescue and firefighting operations. In the survey, 580 professional firefighters participated: all were men aged 20–58 years old (M (mean age) = 35.26 year, SD = 6.74) and came from 12 different provinces. The respondents represented the entire geographical area of central Poland (Masovia and Świętokrzyskie), northern (Pomerania and Kuyavian–Pomerania), south (Lesser Poland, Subcarpathia, Silesia, and Opole), eastern (Lublin) and western (West Pomerania). Approximately 10% of the surveyed firefighters came from each of the mentioned provinces. 

Sociodemographic characteristics of the sample are presented in Table 1, education and work characteristics are presented in Table 2.

### 2.2. Instruments

Sense of self-efficacy was measured using the Polish version of the General Self-Efficacy Scale (GSES), authored by Schwarzer, Jerusalem, and Juczyński [51,52]. The GSES measures the strength of general belief in one’s effectiveness in dealing with difficult situations and obstacles. The scale consists of 10 items that are rated on a 4-point Likert-type scale, ranging from 1 = “not at all true” to 4 = “very true”. The Polish version of the GSES meets psychometric standards; Cronbach’s *α* coefficient was 0.85 [51]. In our study Cronbach’s *α* coefficient was 0.76.

Burnout was measured using the Polish version of the Link Burnout Questionnaire (LBQ) [53,54,55]. The LBQ relates to feelings about one’s professional work. It consists of 24 items that are rated on a 6-point scale (ranging from 1 = “never” to 6 = “every day”). The scale covers four burnout dimensions: 1) psychophysical exhaustion, which describes the subjective state of being exhausted, experience of fatigue, and tension (six items); 2) relationship deterioration, which relates to the quality of relations with patients or clients (six items); 3) sense of professional inefficacy, which is related to evaluation of one’s own professional competences (six items); and 4) disillusion, which relates to disappointment with one’s own achievements and results and constitutes the opposite of job enthusiasm, passion, and satisfaction (six items). The Polish version of the LBQ has satisfactory internal validity and stability, although they differ depending on the tested professions: the weakest values have been shown to be for the sense of professional inefficacy subscale for a sample of therapists [53]. The stability of Polish version is comparable to original one in three subscales: exhaustion, relationship deterioration, and disillusion (*r* = 0.85, 0.86, and 0.85, respectively), and is significantly lower for professional inefficacy subscale (*r* = 0.45). In the standardized Polish sample for the group of uniformed services, the Cronbach’s *α* coefficients were: exhaustion: 0.81; relationship deterioration: 0.73; professional inefficacy: 0.56; and disillusion: 0.85 [53]. In the sample of firefighters, we observed a similar tendency—the lowest Cronbach’s *α* coefficient was for professional inefficacy subscale: 0.60, while for exhaustion, relationship deterioration and disillusion were: 0.82, 0.73, and 0.81, respectively. 

The Perceived Stress Scale (PSS) by Cohen [56] is an instrument for measuring the perception of experienced stress. The Polish version of the scale that was used in this study was developed by Juczyński and Ogińska-Bulik [52]; it consists of 10 items that define the actual level of stress and effectiveness of coping with difficult life events. The items are rated on a 5-point scale (ranging from 0 = “never” to 4 = “very often”). The results made it possible to identify the level of actual experienced stress in the studied group of firefighters. The raw scores were transformed into standard sten scores in which the higher the result, the stronger the perceived stress. Cronbach’s *α* coefficient in a standardized Polish sample was 0.86 [52], in our study Cronbach’s *α* coefficient was 0.82.

### 2.3. Procedure

The study was conducted together with psychologists employed in the State Fire Service from 12 provinces of Poland. They all were trained in study procedure by the first author of the presented study. All participants who agreed to take part in the study completed the Perceived Stress Scale (PSS), the Link Burnout Questionnaire (LBQ), and the General Self-Efficacy Scale (GSES), as well as questionnaires on individual characteristics. The participation was based on a verbal consent. Each participant was informed about the possibility to withdraw from the study at any stage. 

The study protocol was approved by the Bioethics Commission at Jagiellonian University (decision No. 1072.6120.23.2017) and was carried out in accordance with the recommendations of the APA Ethics Code. 

### 2.4. Data Analysis

In the first stage of statistical analysis, the level of self-efficacy, the experienced stress, and burnout among firefighters were estimated. Subsequently, analyses were undertaken to answer the research questions: whether and to what extent self-efficacy moderates the relationship between stress and professional burnout (psychophysical exhaustion, deterioration of relationships with the person who is supported, and disillusion). To test mediation and moderation effects, the approach proposed by Hayes was used [57]. This procedure is recommended for non-normal distributed data, as was observed in the presented study. 

A moderation model was used for statistical analyses to test the influence of variable Mo (moderator) on the relation between the independent variable X and the dependent variable Y; the conditional influence of the independent variable X on the dependent variable Y is described by the formula Y = *b*_1_ + *b*_3_ × Mo, where *b*_1_ is a coefficient describing the relationship between X and Y, and *b*_3_ is a coefficient describing the relationship between the effect of interaction X × M and Y (Figure 1).

## 3. Results

### 3.1. Perceived Stress, Self-Efficacy, and Burnout Characteristics

The characteristics of the level of perceived stress, self-efficacy, and burnout in the sample of firefighters are presented in Table 3.

The analysis of the normality of the distribution was carried out using the Kolmogorow–Smirnow test with the Lilliefors amendment. The distribution of results of all scales differs from normal. The analysis of results in terms of the average values of the severity of the psychophysical state of exhaustion, deterioration in relationships, sense of lack of self-efficacy, and feelings of disillusion indicates that the surveyed firefighters present an average level of severity in all verified scales that assess burnout symptoms. Referring to the norms developed for the Polish population, specifically for the uniformed services [53], the mean value of deterioration of relationships corresponds to the fifth (for relationship deterioration) or the sixth sten (for psychophysical exhaustion, sense of professional inefficacy, and disillusion), which may suggest that the group of studied firefighters compared with other occupations is characterized by an average level of burnout symptoms. 

The analyses may be extended by the context of personal resources, which may be characteristic for employees in a specific profession at the stage of starting a professional career, or during the course of acquired professional experience. Thus, in the second step of our analyses, we show the moderating effect of self-efficacy between stress and burnout in the sample of firefighters.

### 3.2. Perceived Stress and the Sense of Self-Efficacy

Based on the empirical distribution of variables, the level of perceived stress and self-efficacy, the following groups were distinguished: 1) firefighters with a low level of perceived stress, 2) firefighters with a high level of perceived stress, 3) firefighters with a low self-efficacy, and 4) firefighters with a high self-efficacy. The split point was the median value (for the level of perceived stress, *Me* (median value) = 14, for the sense of self-efficacy, *Me* = 31).

The non-parametric Kruskal-Wallis Test results showed that firefighters with 1) a low level of perceived stress (*N* = 293, *M* (mean value) = 147.00) and 2) a high level of perceived stress (*N* = 286, *M* = 436.50) significantly differed in the level of perceived stress (*H* = 434,755, *df* = 1, *p* < 0.001). Similarly, firemen with 3) a low sense of self-efficacy (*N* = 295, *M* = 148.00) and a 4) a high sense of self-efficacy (*N* = 284, *M* = 437.50) significantly differed in the level of self-efficacy (*H* = 441,589, *df* = 1, *p* < 0.001).

### 3.3. Sense of Self-Efficacy and Psychophysical Exhaustion among Firefighters

For exhaustion as the dependent variable, the regression model included the level of perceived stress as the independent variable and self-efficacy as the mediating variable, which explains 27% of psychophysical exhaustion variance *F* (3,574) = 72.33, *p* < 0.001, *R*^2^ = 0.274.

In the model, the influence of perceived stress as the independent variable on exhaustion as the dependent variable is significant (*t* = 3.51, *p* = 0.001, *b*_1_ = 0.92). The relationship between the mediating variable (self-efficacy) and the dependent variable (exhaustion) is not significant (*t* = −0.23, *p* = 0.82). However, the interaction effect of perceived stress and self-efficacy for the dependent variable (exhaustion) is significant (*t* = −1.96, *p* = 0.05, *b*_3_ = −0.02). This result indicates that self-efficacy has a moderating effect on the relation between the level of perceived stress and exhaustion (Figure 2).

The results of the statistical analysis indicate that the sense of self-efficacy changes the direction and strength of the relationship between perceived stress and psychophysical exhaustion (Figure 3).

As shown in Figure 3, a lower sense of self-efficacy is associated with a higher level of psychophysical exhaustion for firefighters experiencing both low and high levels of stress. Therefore, regardless of the level of stress experienced by firefighters (low or high), those who have lower self-efficacy will show a greater tendency to develop psychophysical exhaustion and aggravated burnout.

### 3.4. Sense of Self-Efficacy and Relationship Deterioration among Firefighters

The regression model that takes into account relationship deterioration as the dependent variable, level of perceived stress as the independent variable, and self-efficacy as the intermediate variable, explains only 12% of relationship deterioration variance (*F* (3,574) = 25.13, *p* < 0.001, *R*^2^ = 0.116). Further statistical analyses showed no significant relationship between relationship deterioration and the level of perceived stress (*t* = 1.43, *p* = 0.15), nor between the dependent variable and the interaction of the independent variable and the mediating variable (perceived stress level x sense of self-efficacy) (*t* = −0.66, *p* = 0.50). Self-efficacy does not moderate the relationship between stress and lack of engagement in relationships.

### 3.5. Sense of Self-Efficacy and Sense of Professional Inefficacy among Firefighters

For the sense of professional inefficacy as the dependent variable, the regression model that includes level of perceived stress as the independent variable and self-efficacy as the mediating variable explains 20% of professional inefficacy variance (*F* (3,574) = 15.59, *p* < 0.001). The influence of the independent variable (perceived stress) on the dependent variable (sense of professional inefficacy) is significant, *t* = 2.80, *p* = 0.005, *b*_1_ = 0.61. The relationship between the intermediate variable (sense of self-efficacy) and the dependent variable is not significant (*t* = −1.15, *p* = 0.25). The interaction effect of level of perceived stress and self-efficacy for the dependent variable is significant (*t* = −1.93, *p* = 0.05, *b*_3_ = −0.01). This result may suggest that self-efficacy has a moderating function in relation to a lack of the sense of professional effectiveness (Figure 4).

The sense of self-efficacy changes the direction and strength of the relationship between perceived stress and sense of professional inefficacy (Figure 5).

As shown in Figure 5, a lower sense of self-efficacy is associated with a higher level of feeling of lack of professional effectiveness for firefighters experiencing both low and high levels of stress. As the level of perceived stress increases, the sense of professional failure of the firefighters increases. 

### 3.6. Sense of Self-Efficacy and Disillusion among Firefighters

The regression model features disillusion as the dependent variable, level of perceived stress as the independent variable, and self-efficacy as the moderating variable, which explains 23% of the disillusion variance (*F* (3,574) = 57.10, *p* <0.001). In the model, the influence of perceived stress on disillusion is significant (*t* = 3.55, *p* = 0.004, *b*_1_ = 1.64). The relationship between self-efficacy and disillusion is not significant (*t* = 0.43, *p* = 0.66). The interaction effect of the level of perceived stress and self-efficacy for the dependent variable is significant (*t* = −2.11, *p* = 0.03, *b*_3_ = −0.02). This result may suggest that self-efficacy has a moderating function in relation to disillusion as a burnout symptom (Figure 6).

The sense of self-efficacy changes the direction and strength of the relationship between perceived stress and disappointment with professional work (Figure 7).

The Figure 7 shows that lower self-efficacy is associated with a higher level of disappointment and lack of job satisfaction for firefighters experiencing both low and high levels of stress. With an increase in the level of perceived stress among the studied firefighters, their sense of disappointment with work and lack of satisfaction and passion from the job (disillusion) also increases.

## 4. Discussion

The obtained results indicate that the level of burnout symptoms and the level of perceived stress in the examined 580 firefighters are within the limits of average results. Referring to the norms for different occupations (teachers, nurses, doctors, policemen, and officers of the prison service, and the comparison group consisted of IT workers, engineers, and accountants) presented by Jaworowska [53] in the standardized study, the burnout symptoms in tested firefighters may be determined as average; in each burnout dimension, results correspond with a range of average scores (between fifth and sixth sten) in other occupations [53]. As regards the level of perceived stress among firefighters, similar results were obtained by Dudek [58] in normalization studies conducted by a team at the Institute of Occupational Medicine. The results of the study conducted by Oskwarek and Tokarska-Rodak [59] indicated that the group of 110 firefighters also declared that they did not feel excessive stress in their professional work. In their study, similar sociodemographic indicators were considered as in the presented study (age, gender, marital status, education); however, the respondents in that study were not recruited from such a large area of Poland as the presented analysis. The perceived stress in their study was assessed on the basis of the authors’ questionnaire, in which subjects were asked about the stress they experienced in performing their professional duties (responsibility for the lives of victims and stress symptoms such as accelerated breathing and heart rate, increased pressure, irritability, and fatigue). It is also not possible to directly compare the results of our study with the results of Oskwarka and Tokarska-Rodak’s study because different variable measurement tools were used. Our research used standardized tools that made it possible to determine the general level of firemen’s experienced stress. 

It could be expected that firefighters performing such a demanding and high-risk profession should perceive excessive occupational stress and develop some burnout symptoms. The presented results do not support such intuitive prediction. It may be linked with the specificity of the studied profession. At first, candidates to fire service, due to the general knowledge about firefighter’s responsibilities and demanding tasks, may be more aware of the job requirements and conscious of the risk associated with performing firefighter’s duties. Thus, it is more probable that candidates expecting high stress at work are more resistant to stress in their initial characteristics. Our findings shows that firefighters have higher sense of self-efficacy (results correspond with scores of seventh sten in norms developed by Juczyński [51]). Furthermore, to some extent, firemen’s engagement in their work may be stimulated by learned and trained skills. For example, since they help others as part of their duties, their involvement in relationships with others should be at least average as this is just their job context. Different burnout dimensions: exhaustion, relationship deterioration, professional inefficacy and dissatisfaction with work may depend on personal resources (such as self-efficacy), what requires further research. 

Considering the evaluation of the moderating effect of self-efficacy among firefighters on the relationship between stress and burnout, Ogińska-Bulik and Kaflik-Pieróg [27] confirm similar data. They indicate the importance of the sense of self-efficacy in burnout syndrome among emergency service workers, including firefighters. In the presented study it was also shown that with the increase of the level of experienced stress in the studied firefighters, their sense of disappointment with work and lack of satisfaction (disillusion) also increases. In the study of Ogińska-Bulik and Kaflik-Pieróg [27], the sense of effectiveness among firefighters negatively correlated with emotional exhaustion, while there were no significant relationships between the sense of self-efficacy and other components of occupational burnout. In contrast, slightly different results were obtained in our study: lower self-efficacy was associated with a higher level of psychophysical exhaustion for firefighters experiencing both low and high levels of stress. However, the sample in the previous study consisted of a group of 100 firefighters from only one province [27]. In this study, we present the results of 580 respondents from all over Poland, which might have influenced the obtained results. 

The presented results indicate that the state of exhaustion, the core component of burnout syndrome, is moderated by self-efficacy, both in low- and high-perceived stress conditions. It suggests that irrespective of the level of stress at work, lower levels of self-efficacy evoke stronger feelings of psychophysical exhaustion. This notion may be explained in the context of Conservation of Resources (COR) Theory developed by Hobfoll [60]. In light of this theory, employees with lower personal resources are expected to experience resource losses and what may negatively influence employee’s psychological well-being. In addition, our results showed that the sense of self-efficacy played a moderating function between the level of perceived stress and sense of professional inefficacy. The sense of self-efficacy changed the direction and strength of the relationship between perceived stress and disappointment with professional work (disillusion). It may be hypothesized that the sense of self-efficacy evokes greater job involvement and enthusiasm, which may play a buffering role between a stressful situation and disillusion with work. Faith in one’s own abilities may influence greater belief in one’s own coping strategies in both demanding and and successful situations at work. Thus, the moderating effect between occupational stress and disillusion may be of significant importance in a group of firefighters.

The problem of measuring the moderation of various personal resources in the development of perceived stress and occupational burnout in a group of occupations involving special risks requires further studies. Smith et al. [8] showed that burnout significantly influences safety performance in firefighters. Interestingly, they also emphasized that firefighters are less likely to reveal self-protective behaviors, which may have further consequences for their health and wellbeing. According to Regehr et al. [61], experienced firemen reveal lower self-efficacy compared to recruits, which is associated with traumatic stress and depressive symptoms. Similarly, experienced firemen had a lower level of social support. If such relationships are observed, it is highly important to direct future research in firefighters at analyzing the strength of moderation of such personal resources as self-efficacy, self-esteem, resilience, sense of coherence, dispositional optimism, and experienced social and family support. 

The present study explores the study of Sattler, Boyd, and Kirsch [62], in which the authors pointed out the need to augment firefighter resources and showed support for COR Theory. They emphasized that maintenance and acquisition of individual resources play a substantial role in preventing negative consequences of work stress and critical incidents among firefighters. According to their study, personal characteristic resources are positively associated with post-traumatic growth and negatively with post-traumatic stress symptoms. Prati et al. [16] observed that self-efficacy is an important resource in rescue workers. They showed that in a group of workers with a low level of self-efficacy, stress was significantly associated with professional quality of life. Our findings explicitly show how self-efficacy as an individual resource may influence specific burnout symptoms and how it interacts with perceived stress. The role of personal resources in high risk professions has been indicated in previous studies [16,19,27,61]; the aim of the present study was to test precisely one individual resource and characterize its specific relations with the consequences of work stress; it may help in planning interventions and preventive programs in firefighting services. 

Regarding of the limitations, the present work is a cross-sectional study and is based on self-reports as assessment tools. Although the sense of self-efficacy and perceived stress are naturally based on self-reports, burnout and stress may also be related to other objective measures. Especially nowadays, when neuroimaging and laboratory techniques enable monitoring of more sophisticated indicators of burnout, the use of objective measures would significantly improve the reliability of this research area and explore the character of the relationships between the presented variables. For instance, some objective measures of burnout could be considered, including event-related potentials [38,39], hair cortisol [63], or even functional brain activity [64]. On the other hand, stress levels could be assessed on the basis of real stressful events, as this seems to be easier to monitor and evaluate in firefighting work, especially when compared to other professions such as teaching or management. Additionally, an experimental design and longitudinal study could be introduced to evaluate how sense of self-efficacy influences burnout symptoms. Although this design is sensitive to confounders, it would be able to describe cause and effect relationships.

## 5. Conclusions

The obtained results supported the study hypothesis and answered the research questions. Sense of self-efficacy turned out to be an important moderator of the relationship between the level of perceived stress and the three components of professional burnout among firefighters. Significant interactive effects relate to the sense of self-efficacy in moderating between experienced stress and psychophysical exhaustion, and between sense of professional inefficacy and disillusion. However, the sense of self-efficacy does not moderate between experienced stress and one burnout symptom, i.e., relationship deterioration. This burnout symptom seems to be independent from self-efficacy and in further studies should explore possible moderators or mediators between perceived stress at work and deterioration in relationships with others (e.g., clients, patients, co-workers). In identifying moderating and mediating variables, qualitative research may bring significant input [65]. In future studies, other personal resources that may be crucial for firefighters’ health and satisfaction should be explored, such as self-esteem and happiness [66].

Regardless of the level of stress (low or high) experienced by the studied firefighters, those who have lower self-efficacy show a greater tendency to develop psychophysical exhaustion. For firefighters experiencing both low and high levels of stress, lower self-efficacy is also associated with a higher level of professional inefficacy. As the level of stress increases, the sense of professional inefficacy among firefighters increases. In addition, lower self-efficacy is associated with a higher level of disappointment and a lack of job satisfaction for firefighters experiencing both low and high levels of stress. Together with an increase in the level of perceived stress among studied firefighters, their sense of disillusion also increases. 

In conclusion, self-efficacy as a significant component of the self can act as a factor that modifies the course of mutual dependence between experienced stress and professional burnout among firefighters. The obtained results indicate the role of strengthening the sense of self-efficacy as a factor that counteracts burnout. Thus, special attention should be paid to self-efficacy in burnout prevention, pro-health activities, and psychoeducation among firefighters.

Considering the limitations of the study: cross-sectional research design and self-reported methods used to identify the levels of burnout syndrome, the presented conclusions need further support and should be verified by studies using also objective methods to assess burnout level and possibly longitudinal study design. 

## Figures and Tables

**Figure 1 ijerph-16-00183-f001:**
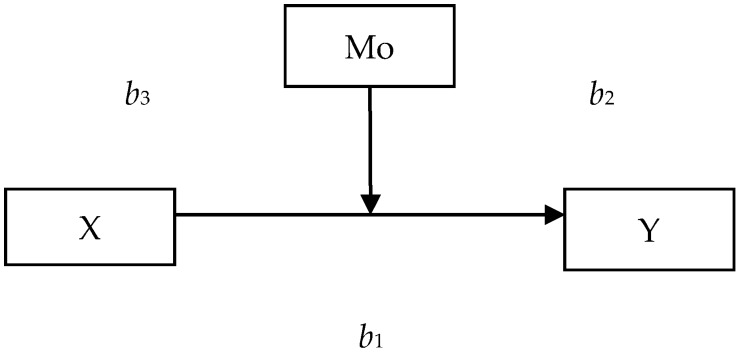
The moderation model; the hypothetical mediation model after Hayes [57], in which variable X (perceived stress) is a predictor, Mo (sense of self-efficacy) is a moderator, and Y (burnout) is a dependent variable. Interaction indexes: 1) *b*_1_ refers to the influence of a predictor on a dependent variable; 2) *b*_2_ refers to the influence of a moderator on a dependent variable; and 3) *b*_3_ refers to interaction effect of a predictor and a moderator for a dependent variable.

**Figure 2 ijerph-16-00183-f002:**
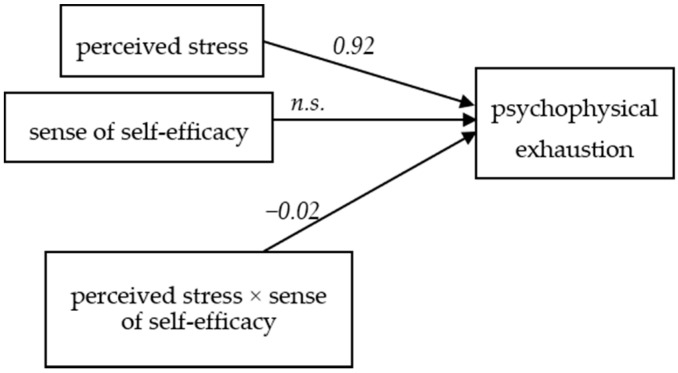
The model of relations between the perceived stress and psychophysical exhaustion, moderated by the sense of self-efficacy (*n.s.*: not significant).

**Figure 3 ijerph-16-00183-f003:**
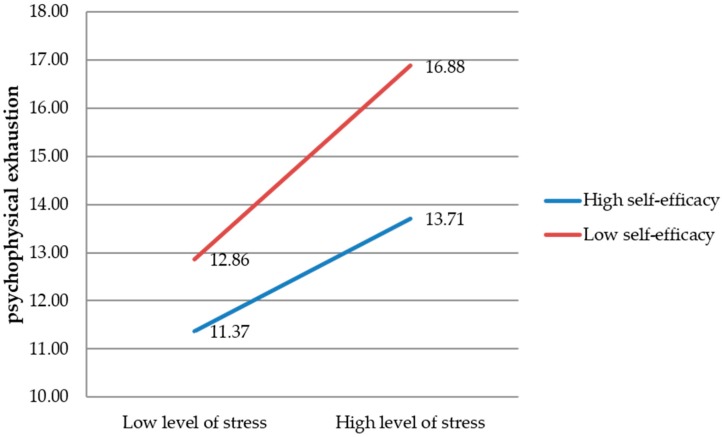
The impact of the sense of self-efficacy on the relationship between the level of perceived stress and psychophysical exhaustion.

**Figure 4 ijerph-16-00183-f004:**
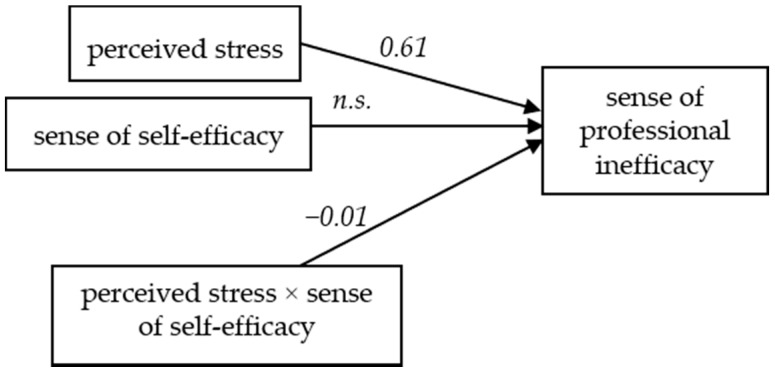
The model of relations between the perceived stress and the sense of professional inefficacy, moderated by the sense of self-efficacy (*n.s.*: not significant).

**Figure 5 ijerph-16-00183-f005:**
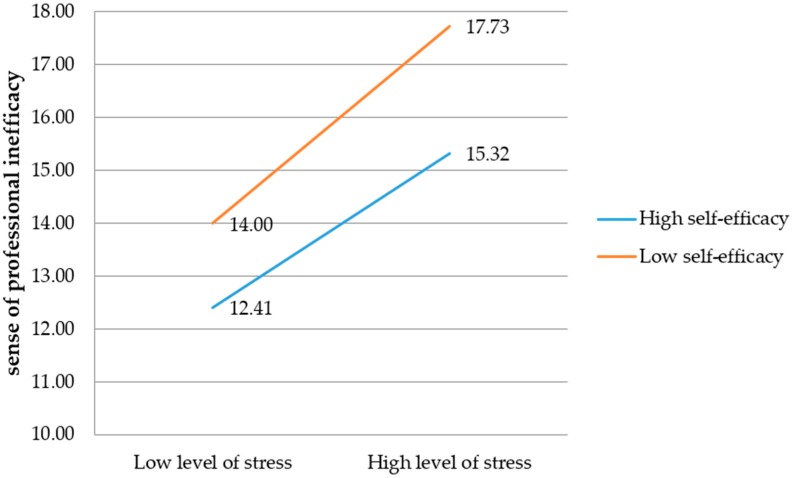
The impact of the sense of self-efficacy on the relationship between the level of perceived stress and the sense of professional inefficacy.

**Figure 6 ijerph-16-00183-f006:**
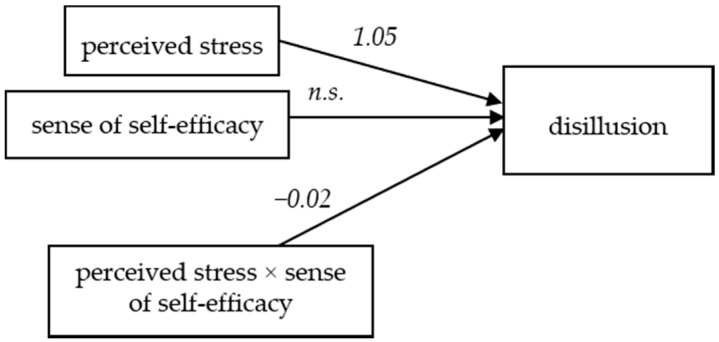
The model of relations between the perceived stress and disillusion, moderated by the sense of self-efficacy (*n.s.*: not significant).

**Figure 7 ijerph-16-00183-f007:**
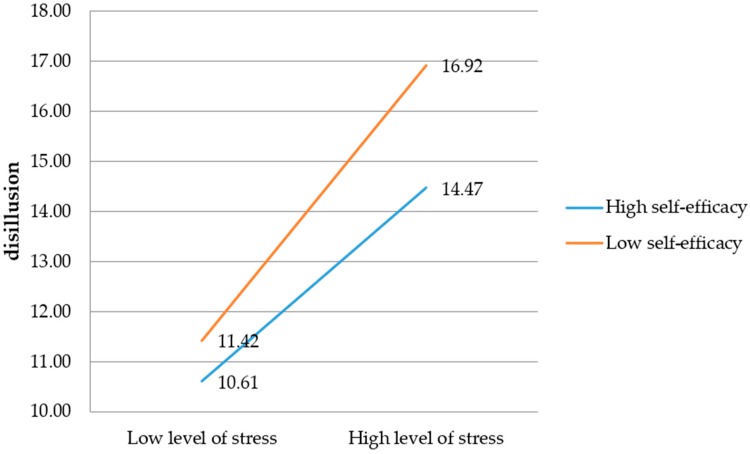
The impact of the sense of self-efficacy on the relationship between the level of perceived stress and disillusion.

**Table 1 ijerph-16-00183-t001:** Sociodemographic characteristics of the sample (*N* = 580).

Sample Characteristic	*N*	%
Gender		
men	580	100
Age		
25 years old or less	43	7.41
26–30 years	104	17.93
31–39 years	288	49.66
40–49 years	133	22.93
over 50 years	12	2.07
Marital status		
single	113	19.48
married	428	73.79
cohabiting	16	2.76
divorced/separated	22	3.79
Children		
0	169	29.14
1	139	23.97
2	226	38.97
3 or more	45	7.76
Place of residence		
countryside	308	53.10
city	272	46.90

Note: data available for Marital status and Children: *N* = 579.

**Table 2 ijerph-16-00183-t002:** Sample characteristics in education and work experience (*N* = 580).

Sample Characteristic	*N*	%
Education		
secondary	291	50.17
post-secondary	9	1.55
higher	280	48.28
Work experience		
3 years or less	45	7.76
4–8 years	152	26.21
9–15 years	238	41.03
16–25 years	130	22.41
over 25 years	11	1.90
Work system		
shift work	541	93.28
day shifts	36	6.21
Professional status		
executives	386	66.55
commanders	193	33.28

Note: data available for Work experience: *N* = 576; Work system: *N* = 577; Professional status: *N* = 579.

**Table 3 ijerph-16-00183-t003:** Descriptive statistics for the Perceived Stress Scale (PSS), the General Self-Efficacy Scale (GSES), and Link Burnout Questionnaire (LBQ) (*N* = 580).

Scales	Mean Value	SD	Min.	Max.	Median	Asymmetry	Kurt	Z	*p*
PSS	14.86	5.72	1	32	14	0.22	−0.07	0.07	<0.001
GSES	31.89	3.58	20	40	31	0.20	0.38	0.13	<0.001
LBQ									
psychophysical exhaustion	15.71	5.61	3	33	15	0.57	−0.08	0.09	<0.001
relationship deterioration	16.55	4.40	4	30	16	0.10	−0.33	0.08	<0.001
sense of professional inefficacy	13.47	4.39	4	32	13	0.47	0.30	0.08	<0.001
disillusion	13.71	6.13	5	34	13	0.68	−0.07	0.10	<0.001

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
