# Peer review of "Self-Efficacy as a Moderator between Stress and Professional Burnout in Firefighters"

_ijerph, 2019, doi:10.3390/ijerph16020183_

Round 1

Reviewer 1 Report

I congratulate the authors for the research carried out. The paper present a research topic which is specially relevant in high risk professions as is the firefighters.

However in the attached document I expressed some doubts and suggestions to improve the global quality of the paper.

Also I recommended some moderation to extract the main conclusions of this research due the research design (cross-sectional) and the methods used to identify the levels of burnout (self-reported).

Best regards,

---

Author Response

Dear Reviewer,

we would like to thank you for your effort to point all weak aspects of our work and for all remarks, which we have found very helpful in improving our manuscript. In attached pdf, we present our responses to all your comments, in the hope that we have been able to satisfy all the raised concerns. As you saggested, in the conclusions we refer to study design and methods used to point the limitations of our work.

With kind regards,

Authors

Reviewer 2 Report

The paper is well done and very interesting, both in theoretiocal and applicative terms. An additional strenght refers to the sample.

I sugest authors should change the title, deleting "sense"...self efficacy is simply self-efficacy...of course also along the paper, I'd delete "sense" and prefer the simple "self efficacy".

I suggest authors should use instead of the ANOVA in the paragraph 3.2 the relative non-parametric test for means analysis, sinece they declare that the dataset shows problems in terms of normality.

In the paragraph 3.6 authors talk about self-efficacy as a mediating variable but actually it is a moderator one.

they should add some limitations and future research agenda; in particular, considering future research agenda I would see space for studying additional mediating variables in order to enrich the causal relationship they have forund. for example, it would be interesting to study also positive variables like for example happinness that along with self-esteem may play a role on health and satisfaction as already found in other contexts (see Benevene et al., 2018). There would be space also for studying the same variables but with different types of analysis, adopting qualitative methods

I suggest authors should enlarge and refresh  their reference list with the following studies

Benevene, P., Ittan, M. M., & Cortini, M. (2018). Self-esteem and happiness as predictors of school teachers' health: The mediating role of job satisfaction. Frontiers in Psychology, 9(JUL) doi:10.3389/fpsyg.2018.00933

Duran, F., Woodhams, J., & Bishopp, D. (2018). An interview study of the experiences of firefighters in regard to psychological contract and stressors. Employee Responsibilities and Rights Journal, 30(3), 203-226. doi:10.1007/s10672-018-9314-z

Author Response

Dear Reviewer,

Thank you very much for the review and all your comments. They helped us a lot to improve our work. We deleted “the sense of…” in many places – leaving in some to be consistent with the literature we refer to. But we agree that it is easier to read the text if it is in a concise form.

In the paragraph 3.2 we replaced the ANOVA with the non parametric Kruskal Wallis Test.  

In the paragraph 3.6 we corrected mediating variable into moderating variable (thank you for pointing it!).

In conclusions, we added some limitations and future research agenda, according to your suggestions. This new part of the test is in lines: 480-485. We also refreshed the references with studies you proposed – thank you for your help.

With kind regards,

Authors

Reviewer 3 Report

The topic of the reviewed paper relates to the scope of International Journal of Environmental Research and Public Health. Although the objectives of the study were previously addressed in the literature, a survey on such scale taking into account recruits from large area of the country were not previously undertaken. The paper reads really nicely, the objective is clearly stated methods are proper and well described which guarantees reproducibility of the study. The results are presented clearly and appropriate statistical methods are used to analyze the data. Discussion of presented results is carried out in relation to the previously published literature. One suggestion for the authors would be to add the information on the internal validity and stability of LBQ, perhaps it would be possible to give a range of values for tested professionals for specified burnout dimensins.

Author Response

Dear Reviewer,

Thank you very much for the review and all your comments. We added information about psychometric characteristics of methods used in the study. A range of values for tested professionals for specified burnout dimensions is presented in Table 3.

Thank you for your supportive review – it gives the motivation to develop our work in this field!

With kind regards,

Authors

Reviewer 4 Report

The aim of the study is very interesting, reporting about an important field of occupational health such as firefighting, deepening the concept of ‘sense of self-efficacy’. Each part of the manuscript is good. Therefore, I have no remarks to make.

Author Response

Dear Reviewer,

Thank you very much for your review and your appreciation to our work.

It is very supportive and  motivates us to further efforts.

With kind regards,

Authors